# The Characteristics and Prognosis of Alpha-Fetoprotein and Des-Gamma-Carboxy Prothrombin Double-Negative Hepatocellular Carcinoma at Baseline in Higher BCLC Stages

**DOI:** 10.3390/cancers15020390

**Published:** 2023-01-06

**Authors:** Takakazu Nagahara, Takaaki Sugihara, Takuya Kihara, Suguru Ikeda, Yoshiki Hoshino, Yukako Matsuki, Takuki Sakaguchi, Hiroki Kurumi, Takumi Onoyama, Tomoaki Takata, Tomomitsu Matono, Naoyuki Yamaguchi, Hajime Isomoto

**Affiliations:** 1Division of Gastroenterology and Nephrology, Department of Multidisciplinary Internal Medicine, Faculty of Medicine, Tottori University, Yonago 683-8504, Japan; 2Department of Gastroenterology, St. Mary’s Hospital, Himeji 670-0801, Japan; 3Department of Endoscopy, Nagasaki University Hospital, Nagasaki 850-8501, Japan

**Keywords:** hepatocellular carcinoma, alpha-fetoprotein, des-gamma-carboxyprothrombin, survival

## Abstract

**Simple Summary:**

In clinical settings, some cases with hepatocellular carcinoma (HCC) demonstrate negativity in both alpha-fetoprotein (AFP) and des-gamma-carboxyprothrombin (DCP). Most are small and early-stage hepatocellular carcinomas (HCCs). This study aimed to investigate the characteristics and prognosis of AFP (<20 ng/mL) and DCP (<40 mAU/ml) double-negative HCC (DNHC) in higher BCLC stages. We confirmed that 120 of 374 patients (32.1%) were DNHC, and 17 (14.7%) were in higher stages (BCLC-B, C, and D). In higher-stage HCC, there was no difference in BCLC staging; however, there were significantly more cases under TNM Stage III in DNHC (71.0% vs. 41.4%, *p* = 0.026). This is due to the tumor size, which can influence treatment. Curative locoregional therapy was dominantly applied in DNHC (*p* = 0.022). Therefore, survival was significantly better in DNHC (*p* = 0.027).

**Abstract:**

Alpha-fetoprotein (AFP) and des-gamma-carboxyprothrombin (DCP) are widely used as tumor markers to diagnose hepatocellular carcinoma (HCC). Some advanced HCCs demonstrate neither AFP nor DCP. This study investigated the characteristics and prognosis of AFP (<20 ng/mL) and DCP (<40 mAU/ml) double-negative HCC (DNHC) in higher-stage HCC. Between April 2012 and March 2022, 419 consecutive patients were enrolled with newly diagnosed HCC and 372 patients were selected that were diagnosed by histopathology and/or imaging. AFP-negative, DCP-negative, and double-negative HCC were identified in 262 patients (70.4%), 143 patients (38.2%), and 120 patients (32.3%), respectively. In higher-BCLC stages (BCLC-B, C, and D), 17 patients (14.7%) were DNHC. Although there was no difference in BCLC staging, there were more cases under TNM Stage III in DNHC (71.0% vs. 41.4%, *p* = 0.026). The median maximum tumor diameter was smaller in DNHC [3.2 (1.8–5.0) vs. 5.5 (3.5–9.0) cm, *p* = 0.001] and their median survival time was significantly better, even in higher-stage HCC [47.0 (24.0–84.0) vs. 19.0 (14.0–30.0) months, *p* = 0.027). DNHC in higher-BCLC stage HCC is independent of BCLC staging, characterized by a tumor diameter < 5 cm, and is treatable with a good prognosis.

## 1. Introduction

Primary liver cancer was the third leading cause of cancer death worldwide in 2020 with approximately 830,000 deaths according to GLOBOCAN 2020 data [1]. Hepatocellular carcinoma (HCC) comprises 75–85% of primary liver cancers. Alpha-fetoprotein (AFP) is the most used serum biomarker for diagnosing and monitoring HCC. Bergstrand and Czar identified AFP as a new fraction of alpha globulins in human fetal serum in 1956 [2]. Abelev discovered the relationship between AFP and hepatoma in 1963 [3]. In 1964, Tatarinov reported the first case in which the serum AFP levels were elevated in the sera of a liver cancer patient [4]. However, not all HCCs secrete AFP, and 40–46% of HCC patients reportedly remain AFP-negative (<20 ng/mL) [5,6,7]. 

A large-scale study demonstrated that an AFP-negative (<20 ng/mL) rate was found in 52.2% (262/502) of patients with small HCCs (<3 cm) and in 53.5% (51/95) of patients at TNM Stage I [6]. In addition, AFP may be elevated in cases of cirrhosis or hepatitis [8,9,10]. Therefore, AFP has a low specificity [11,12]. The optimal cut-off is still controversial [13,14]. In 1984, Liebman et al. reported an elevated plasma des-gamma-carboxyprothrombin (DCP), also known as a prothrombin molecule that is induced by vitamin K absence (PIVKA II), in 67% of patients with HCC [15]. Subsequently, other researchers reported that DCP is elevated in around 50–60% of patients with HCC and 15–30% of patients with HCC < 3 cm [16,17,18,19]. DCP is reportedly associated with portal vein invasion and poor prognosis [20,21]. Moreover, lens culinaris agglutinin-reactive fraction of AFP (AFP-L3) is now available to complement or replace AFP in diagnosing HCC [22,23]. It is particularly useful in early HCC < 2 cm because of its high specificity [24]. 

AFP and DCP are independent of one another [21]. Therefore, the combination of AFP and DCP may improve detection rates [25,26,27,28,29,30]. However, there are cases of AFP- and DCP-negative HCCs. Pan et al. recently reported a large-scale analysis of prognostic values of AFP and DCP in HCC and found that 12.9% of HCCs were both AFP- and DCP-negative (defined as AFP < 25 ng/mL and DCP < 40 mAU/mL) [31]. In the study, 57.7% (218/378) were BCLC-A in the AFP- and DCP-negative group. Miyaaki et al. indicated 35% (39/110) were AFP-L3 and DCP double-negative, and Toyoda et al. found 23.3% (159/685) were AFP, AFP-L3, and DCP triple-negative HCCs which were smaller than the others [32,33]. From these reports, the biomarker-negative cases consist of relatively small, early-stage HCC. These cases are the “upcoming HCCs” that cannot produce biomarkers because they are still small and in the earlier clinical stages. There may be a biomarker sensitivity issue and a lead time effect. However, we have experienced some advanced HCCs that can produce biomarkers that demonstrate neither AFP nor DCP. AFP and DCP double-negative HCC (DNHC) in higher stages may differ from DNHC in the earlier stages. Here, we aimed to investigate the characteristics and prognosis of DNHC in the higher stages.

## 2. Patients and Methods

This was a single-center, retrospective, observational study involving 417 patients from our hospital that were diagnosed with HCC that examined AFP and DCP at baseline levels between April 2012 and March 2022. Patients who took warfarin, vitamin K, or antibiotics were excluded because they would influence the DCP levels. The diagnosis was based on histopathology in 172 patients. Among the patients whose histopathology was unavailable, we selected only those that were diagnosed with typical HCC according to LR ≥ 4 (arterial phase hyperenhancement) on the CT/MRI Liver Imaging Reporting and Data System LI-RADS) v2018 [34]. Finally, there were 372 eligible patients (Figure 1). In this study, higher-stage HCC included the intermediate (B), advanced (C), and terminal (D) Barcelona Clinic Liver Cancer (BCLC) stages [35]. TNM was staged according to the UICC eighth edition [36]. DNHC was defined as both AFP-negative (<20 ng/mL) and DCP-negative (<40 mAU/ml). 

### Statistical Analysis

Welch’s *t*-test or chi-square test was applied to compare the two independent groups. Spearman’s correlation coefficient (shown as rS) was used for evaluating the correlation between two variables. The cumulative survival rate was calculated by the Kaplan–Meier method, and significant differences between the two groups were calculated using the log-rank test. Cox proportional hazards regression analysis was applied to evaluate the survival factors. All statistical tests were performed using StatFlex (Windows ver. 7.0; Artech, Osaka, Japan). Values are expressed as the mean ± SD or median (interquartile range). Median survival time (MST) (95% confidence interval) was used for the survival analysis. Statistical significance was set at *p* < 0.05.

## 3. Results

### 3.1. Patient Characteristics 

Patient characteristics are shown in Table 1. We enrolled 372 patients [283 men, 89 women, 71.4 ± 10.4 years] in this study. The causes of their liver diseases included the hepatitis B virus (HBV) (*n* = 83), hepatitis C virus (HCV) (*n* = 96), HBV+HCV (*n* = 2), alcohol (*n* = 110), non-alcoholic fatty liver disease (NAFLD)(*n* = 18), and other causes (*n* = 63). A total of 199 patients had chronic hepatitis and 173 had cirrhosis. There were 115, 51, and 7 patients that were classified into Child–Pugh classes A, B, and C, respectively. There were 92, 164, 52, 54, and 10 patients that were classified into BCLC-0, -A, -B, -C, and -D, respectively. A total of 304 cases were classified as typical HCC (10 LR-4 and 354 LR-5) by enhanced CT/MRI. Of the patients who had a biopsy or resection, there were 58, 103, and 11 cases of well-, moderate-, and poorly-differentiated HCC, respectively. A total of 23 patients (6.2%) were AFP single-positive, 142 (38.2%) were DCP single-positive patients, and 87 (23.4%) were double-positive (DPHC) patients. The median duration of follow-up was 32.0 (13.0–59.0) months.

A total of 262 patients (70.4%) were AFP-negative HCC (ANHC) and 143 patients (38.2%) were DCP-negative HCC (DCPNHC). There were 120 patients (32.3%) that were DNHC. In higher-stage HCC (*n* = 116), 50 patients (43.1%) were ANH, 20 patients (17.2%) were DCPNHC, and 17 patients (14.7%) were DNHC (Figure 2). 

### 3.2. Characteristics of Patients with DNHC

Patients with DNHC were more likely to have well-differentiated HCC (51.9 vs. 25.8%, *p* = 0.010), very early and early stage (BCLC-0/A) (82.5 vs. 61.5%, *p* < 0.001), and lower TNM stages (IA/B and II) (95.8% vs. 75.4%, *p* < 0.001) compared to other groups (Table 2). 

### 3.3. DNHC in Higher-BCLC Stage HCC

A total of 116 patients were classified as higher-BCLC stage HCC (BCLC-B, C, and D). In higher-stage HCC, 17 patients (14.7%) (10, 6, and 1 in BCLC-B, C, and D, respectively) were DNHC. Comparing DNHC patients with others in the higher-stage HCC, well-differentiated HCC was relatively higher in DNHC (*p* = 0.090). There was no difference in BCLC staging, the Up to 7 criteria, and Kinki criteria; however, there were more cases under TNM Stage III in DNHC (71.0% vs. 41.4%, *p* = 0.026). There was no statistical difference in the vascular invasion between DNHC and the other groups. The median maximum tumor diameter of DNHC was smaller than the other tumors [3.2 (1.8–5.0) vs. 5.5 (3.5–9.0) cm, *p* = 0.001] (Figure 3). Therefore, locoregional therapy [radiofrequency ablation (RFA), resection, trans-arterial chemoembolization (TACE), and stereotactic body radio therapy (SBRT)] was selected more frequently in the DNHC patients (*p* = 0.022). Patients with DNHC were more likely to achieve a complete response to the first treatment (*p* = 0.021) (Table 3).

### 3.4. Survival Analysis

We analyzed the survival of the whole dataset according to the BCLC staging. Survival was well stratified by the BCLC staging [ 0 vs. A, *p* < 0.001; A vs. B, *p* < 0.001; B vs. C, *p* = 0.022]; however, there was no statistical difference between BCLC-C and D (Figure 4a). The one, three, and five -year overall survival (OS) rates of each stage were 97.7%, 92.3%, and 76.5%, respectively, in BCLC-0; 91.0%, 72.9%, and 40.1%, respectively, in BCLC-A; 82.3%, 41.7%, and 24.0%, respectively, in BCLC-B; 56.9%, 29.4%, and 16.6%, respectively, in BCLC-C; and 40.5%, 0.0%, and 0.0%, respectively, in BCLC-D. 

The MST of patients with ANHC was significantly better than the other groups [73.0 (67.0–87.0) vs. 24.0 (16.0–34.0) months, respectively, *p* < 0.001) (Figure 4b). The MST of the patients with DCPNHC was significantly better than the other groups [87.0 (71.0–not reached) vs. 42.0 (34.0–51.0) months, respectively, *p* < 0.001] (Figure 4c). The MST of patients with DNHC was significantly better than the other groups [87.0 (71.0–not reached) vs. 45.0 (35.0–57.0) months, respectively, *p* < 0.001) (Figure 4d). The one, three, and five -year overall survival (OS) rates were 96.6%, 84.3%, and 72.1%, respectively, for DNHC and 80.2%, 56.5%, and 39.6%, respectively, for the other groups. There was no difference between DNHC and AFP single-positive cases. DNHC demonstrated a significantly better prognosis than DCP single-positive cases [87.0 (71.0–not reached) vs. 66.0 (46.0–78.0) months, respectively, *p* < 0.001) and DPHC [19.0 (13.0–27.0) months, *p* < 0.001) (Figure 4e).

In higher-stage HCC (BCLC-B, C, and D), the MST of the patients with ANHC was significantly better than the other groups [39.0 (32.0–55.0) vs. 14.0 (8.0–19.0) months, respectively, *p* < 0.001) (Figure 5a). The MST of the patients with DCPNHC was not statistically different from the other patient groups [42.0 (19.0–55.0) vs. 20.0 (14.0–30.0) months, respectively, *p* = 0.087) (Figure 5b). The MST of the patients with DNHC was significantly better than the other groups [47.0 (24.0–84.0) vs. 19.0 (14.0–30.0) months, respectively, p = 0.027) (Figure 5c). The one, three, and five -year OS rates were 94.1%, 66.8%, and 22.9%, respectively, for the DNHC group and 63.3%, 26.7%, and 18.9%, respectively, for the other groups.

We then performed a Cox proportional hazards regression analysis, including age, sex, previously proven factors that are associated with HCC survival (Up to 7 criteria IN, Child–Pugh scores, and BCLC stage), and DNHC. The Kinki criteria was excluded because the Up to 7 criteria and Child–Pugh scores were already included in the criteria. Age, BCLC-stage, and Child–Pugh scores increased the relative risk for the survival of higher-stage HCC. The Up to 7 criteria IN and DNHC decreased the RR (Table 4) (Figure 6). DNHC was also an independent factor that was associated with the survival of higher-stage HCC.

## 4. Discussion

This study demonstrated that the incidence of DNHC was 14.7% in higher-stage HCC. In higher-stage HCC, there was no difference in BCLC staging between DNHC and the other groups. However, well-differentiated HCC was relatively higher and there were significantly more cases under TNM Stage III in DNHC, which affected the treatment choice. Locoregional therapy (RFA, Resection, TACE, and SBRT) was selected in cases of DNHC. Therefore, survival was significantly better in DNHC, even in higher-stage HCC.

It has long been reported that ANHC (<20 ng/mL) is a distinct entity of lower Edmondson–Steiner grade (Stage I and II) and demonstrates a favorable long-term prognosis [6,7,37]. An experimental study demonstrated that AFP has a regulatory role in angiogenesis and cell invasion during liver cancer development. AFP is actively involved in tumor progression [38]. ANHC (<20 ng/mL) is found in 30%-40% of advanced (TNM Stage III, IV) HCC patients [6, 7]. In our study, the incidence of ANHC in advanced HCC was consistent with other reports (31.3 % in TNM stage ≥ III and 32.3% in BCLC-C and D). The prognosis was significantly better than the other groups. In the molecular-targeted therapies for advanced HCC, ramucirumab is the only therapy that is defined by the baseline AFP level (≥400 ng/mL) according to the REACH2 trial [39]. Cases with AFP < 400 ng/mL originally had a better prognosis, which may be why ramucirumab could not improve OS compared with the placebo in the REACH trial [40]; locoregional therapy may be applied in these cases. DCPNHC was found in 34–64.9% of cases, and the recurrence-free survival and prognosis were better than in patients with higher DCP [18,19,20,21,33,41]. In our study, the incidence and prognosis were consistent with other reports. 

Pan et al. indicated that DNHC had the best OS [31]. However, their data contained data on early-stage tumors; therefore, the “lead time effect” is inevitable. Interestingly, in our study, DNHC still demonstrates a better prognosis than the others, even in higher-stage HCC. The difference is not induced only by the “lead time effect.” Our results indicate that DNHC in higher-stage HCC had a relatively good histological appearance and was under TNM Stage III. It implies HCC heterogeneity in BCLC-B, C, and D. Kudo et al. proposed the “Kinki Criteria” for the subclassification of BCLC-B [42]. They classified BCLC-B into B1 to B3 by CP scoring, Milan criteria, and Up-to-7 criteria. However, in our study, there were no statistical differences in the BCLC-B subgroups. Differences seem to originate from the maximum tumor diameter. The diverging point would be 5 cm as T2 and T3 are divided by 5 cm in TNM staging [36]. The maximum tumor diameter was moderately correlated with DCP levels rather than AFP. Many reports have demonstrated that the tumor diameter increases DCP-positive rates [21,33,43]. Nakamura et al. indicated that the ROC area of DCP was significantly larger than that of AFP in tumors that were greater than 5 cm in diameter [44]. Tsugawa et al. also reported that DCP has predictive power for tumors > 5 cm [45]. Pan et al. also demonstrated that most of their DNHC (defined as AFP < 25 ng/mL and DCP < 40 mAU/mL) were smaller than 5 cm [31]. 

Multivariate analysis demonstrated DNHC is an independent factor for OS of higher-stage HCC. Tumor marker levels are not included in any criteria. However, our study implies that considering the “negativity” of tumor markers should be beneficial in determining treatment and is closely related to the survival of higher-stage HCCs.

In the clinical practice guidelines of HCC by the European Association for the Study of the Liver (EASL), the American Association for the Study of Liver Diseases (AASLD), and the Asian Pacific Association for the Study of the Liver (APASL), biomarkers were mainly discussed in the vein of early detection [46,47,48]. The prognostic aspect of the biomarkers was not mentioned. These guidelines indicated the limited ability of the AFP for early detection of HCC and discussed how to increase the accuracy for detection. They also introduced some novel biomarkers, such as glypican 3 [49], Golgi protein 73 [50], osteopontin [51], circulating cell-free DNA [52], and microRNAs [53]. These markers are under active investigation; however, none have been approved for clinical use. These novel biomarkers might be useful for DNHC in higher-stage HCC.

The difficulty in defining biomarkers that are specific for HCC cells is due to its complex genomic landscape with extensive intratumor and inter-tumor heterogeneities. Meanwhile, an emerging concept is that an interplay between viral infection and host genetic background is crucial for maintaining virome homeostasis or causing human disease [54]. Lui et al. demonstrated how viral infection history, obtained using human blood samples and VirScan analysis of antiviral antibodies, can be used to detect HCC in at-risk patients prior to clinical cancer diagnoses [55]. However, it cannot be applied to increasing non-viral HCC. The challenge of biomarker discovery continues.

Our study has some limitations. It was a small-size, single-center, retrospective observational study. A future large-scale prospective study is warranted. A biopsy was performed in only half of the patients. However, noninvasive imaging plays a key role in the diagnostic and therapeutic strategy for HCC. LI-RADS, used in this study, demonstrated excellent diagnostic accuracy for HCC in a systematic review [56]. Moreover, we only included cases with arterial phase hyperenhancement to achieve higher accuracy. 

## 5. Conclusions

In this study, our results demonstrate that DNHC is a distinct entity that is independent of BCLC staging and may provide a better prognosis at any stage. DNHC in higher-stage HCC was smaller and curative locoregional therapy could be applied, resulting in a better prognosis.

## Figures and Tables

**Figure 1 cancers-15-00390-f001:**
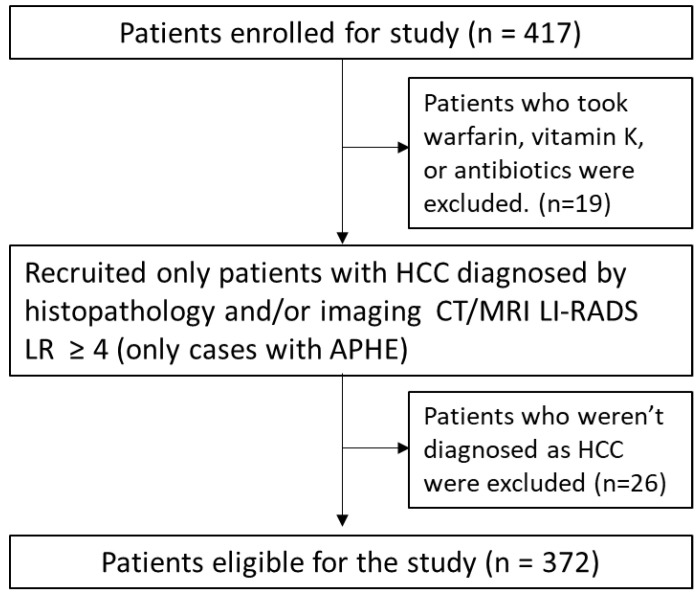
Flowchart of patient selection. Initially, 417 patients with HCC were enrolled, but 19 were excluded for taking warfarin or antibiotics. Only patients with HCC that was diagnosed by histopathology and/or imaging were selected, and 26 patients were excluded. Finally, 372 patients were selected for this study. LI-RADS, Liver Imaging Reporting and Data System; APHE, arterial phase hyperenhancement.

**Figure 2 cancers-15-00390-f002:**
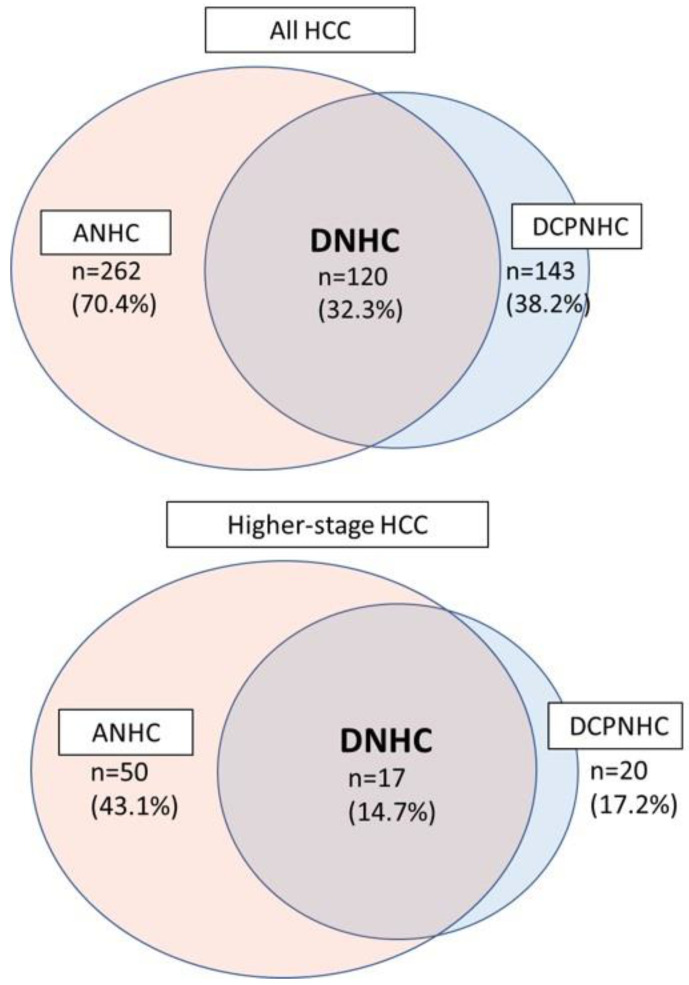
HCC tumor marker negativity. Of all the patients with HCC (*n* = 372), 262 (70.4%) were ANHC, and 143 (38.2%) were DCPNHC. DNHC included 120 patients (32.3%). In higher-stage HCC patients (*n* = 116), 50 patients (43.1%) were ANHC, 20 patients (17.2%) were DCPNHC, and 17 patients (14.7%) were DNHC.

**Figure 3 cancers-15-00390-f003:**
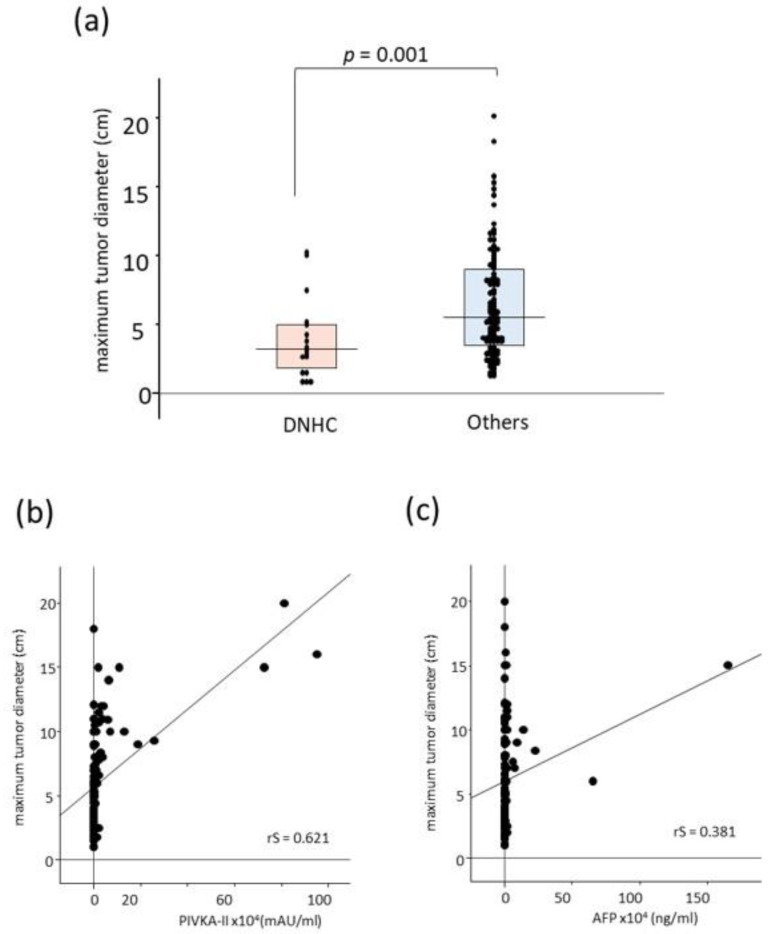
The difference in the maximum tumor diameter and correlation with tumor markers in higher-stage HCCs. (**a**) The median maximum tumor diameter of DNHC was smaller than the other tumors [3.2 (1.8–5.0) vs. 5.5 (3.5–9.0) cm, respectively, *p* = 0.001]. Tumor size was moderately correlated with (**b**) DCP levels (rS = 0.621) rather than (**c**) AFP (rS = 0.381). AFP, alpha-fetoprotein; DCP, des-gamma-carboxyprothrombin; DNHC, double negative HCC; HCC, hepatocellular carcinoma.

**Figure 4 cancers-15-00390-f004:**
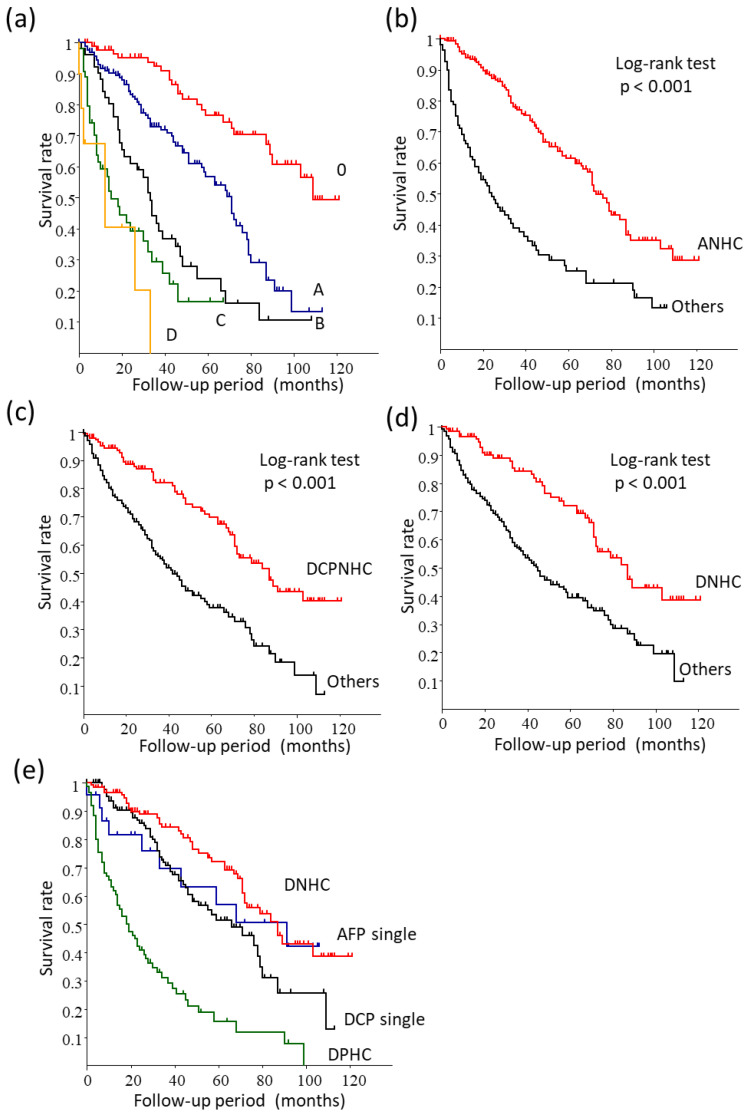
Survival analysis. (**a**) BCLC staging for all patients. It was well stratified (inter-group statistical differences are *p* < 0.001, except for B vs. C; *p* = 0.022, B vs. C; *p* = 0.028, B vs. D; *p* = 0.005, and C vs. D; n.s.). (**b**) Comparison between ANHC and the other patient groups. The survival rate of the patients with ANHC was significantly better than the other patients (*p* < 0.001). (**c**) Comparison of DPCNHC and the other patient groups. The survival rate of the patients with DCPNHC was significantly better than the other patients (*p* < 0.001). (**d**) Comparison between DNHC and the other patients. The survival rate of the patients with DNHC was significantly better than the other patients (*p* < 0.001). (**e**) There was no difference between DNHC and AFP single-positive cases. DNHC cases demonstrated a substantially better prognosis than DCP single-positive cases (*p* < 0.001) and DPHC cases (*p* < 0.001).

**Figure 5 cancers-15-00390-f005:**
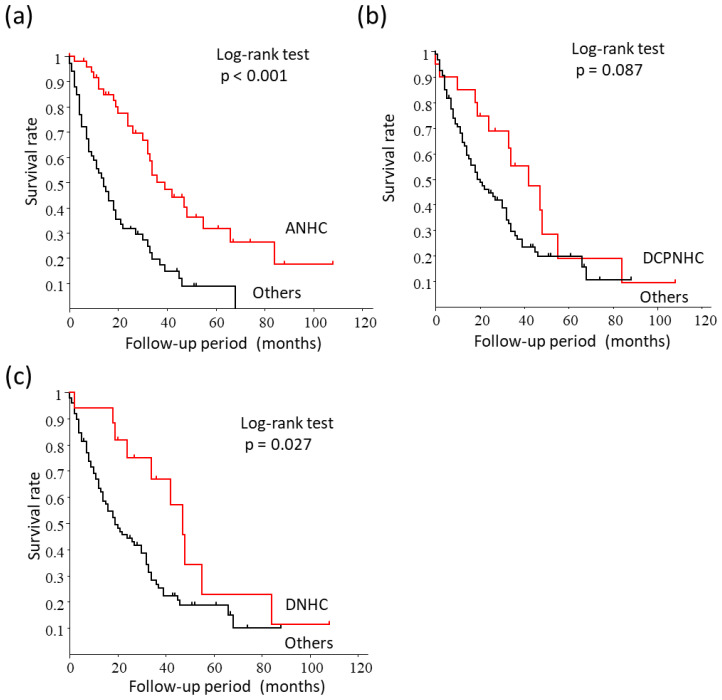
Survival analysis in higher-stage HCC (BCLC-B, C, and D). (**a**) Comparison between ANHC and the other patient groups. The survival rate of the patients with ANHC was significantly better than the other groups (*p* < 0.001). (**b**) Comparison between DCPNHC and the other groups. There was no difference in survival between DCPNHC and the other patient groups (*p* = 0.087). (**c**) Comparison between DNHC and the other patients. The survival rate of the patients with DNHC was significantly better than the other groups (*p* = 0.028).

**Figure 6 cancers-15-00390-f006:**
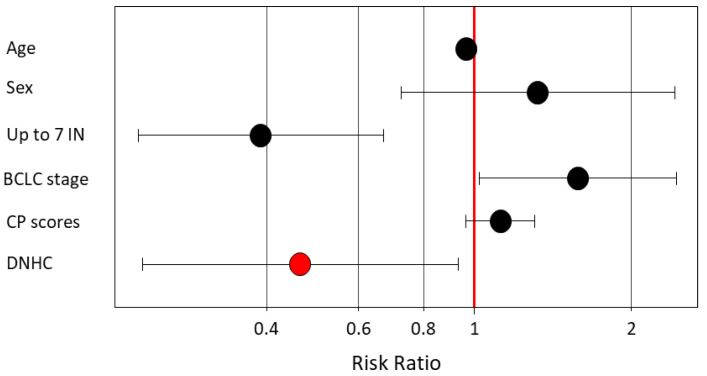
Multivariate analysis for factors that were associated with the OS of higher-stage HCC. Age, BCLC-stage, and Child–Pugh scores increased RR. Up to 7 criteria IN and DNHC decreased the RR. BCLC, Barcelona Clinic Liver Cancer; CP, Child–Pugh; DNHC, double negative HCC.

**Table 1 cancers-15-00390-t001:** Patient characteristics.

Characteristics	*n* = 372	
Male/Female	283: 89	
Age (years)	71.4 ± 10.4	
Etiology		
HBV infection	83	(22.3)
HCV infection	96	(25.8)
HBV+HCV	2	(0.5)
Alcohol	110	(29.4)
NAFLD	18	(4.8)
Others *	63	(16.8)
Liver status		
Cirrhosis	173	(46.5)
CP-A	115	(66.5)
CP-B	51	(29.5)
CP-C	7	(4.0)
Maximum tumor diameter (cm)	2.7 (1.8–4.5)	
Tumor markers		
AFP single-positive	23	(6.2)
DCP single-positive	142	(38.2)
Double-positive (DPHC)	87	(23.4)
Imaging diagnosis		
Typical HCC **	364	(97.8)
Histologically proven HCC	172	(46.2)
Well-differentiated	58	(33.7)
Moderately-differentiated	103	(59.9)
Poorly-differentiated	11	(6.4)
BCLC staging		
0	92	(24.7)
A	164	(44.1)
B	52	(14.0)
C	54	(14.5)
D	10	(2.7)
TNM staging		
IA	91	(24.5)
IB	123	(33.0)
II	91	(24.5)
IIIA	32	(8.6)
IIIB	14	(3.8)
IVA	2	(0.5)
IVB	19	(5.1)
Follow-up duration	32 (13–59) months	

HBV, hepatitis B virus; HCV, hepatitis C virus; CP, Child–Pugh classification; AFP, alpha-fetoprotein; DCP, des-gamma-carboxyprothrombin; HCC, hepatocellular carcinoma; BCLC, Barcelona Clinic Liver Cancer; Data expressed as median (interquartile range) or mean ± SD. Numbers in parentheses refer to the percentage of patients. * Others include autoimmune hepatitis, primary biliary cholangitis, and cryptogenic. ** Defined as LR ≥ 4 (only cases with APHE) on the CT/MRI LI-RADS v2018.

**Table 2 cancers-15-00390-t002:** Comparison between patients with DNHC and other groups.

	DNHC (*n* = 120)		Other (*n* = 252)		*p*-Value
Male/Female	93: 27		190: 62		0.627
Age (years)	70.7 ± 8.9		71.7 ± 11.1		0.452
Etiology					
HBV infection	32	(26.7)	51	(20.2)	0.480
HCV infection	34	(28.3)	62	(24.6)	
HBV+HCV	1	(0.8)	1	(0.4)	
Alcohol	35	(29.2)	75	(29.8)	
NAFLD	5	(4.2)	13	(5.2)	
Others *	13	(10.8)	50	(19.8)	
Liver status					
Cirrhosis	60	(50.0)	113	(44.8)	0.318
CP-A	43	(71.7)	72	(63.7)	0.248
CP-B	16	(26.7)	35	(31.0)	
CP-C	1	(0.2)	6	(5.3)	
Histologically proven HCC	52	(43.3)	120	(47.2)	
Well-differentiated	27	(51.9)	31	(25.8)	0.010
Moderately-differentiated	23	(44.2)	80	(66.7)	
Poorly-differentiated	2	(1.7)	9	(7.5)	
BCLC staging					
0	54	(45.0)	38	(15.1)	
A	49	(40.8)	115	(45.6)	
B	10	(8.4)	42	(16.7)	<0.001
C	6	(5.0)	48	(19)	
D	1	(0.8)	9	(3.6)	
TNM staging					
IA	52	(43.3)	39	(15.5)	
IB	37	(30.8)	86	(34.1)	
II	26	(21.7)	65	(25.8)	
IIIA	4	(3.3)	28	(11.1)	<0.001
IIIB	0	(0.0)	14	(5.6)	
IVA	0	(0.0)	2	(0.8)	
IVB	1	(0.8)	18	(7.1)	

HBV, hepatitis B virus; HCV, hepatitis C virus; CP, Child–Pugh classification; AFP, alpha-fetoprotein; DCP, des-gamma-carboxyprothrombin; HCC, hepatocellular carcinoma; BCLC, Barcelona Clinic Liver Cancer; Data expressed as mean ± SD. Numbers in parentheses refer to the percentage of patients. * Others include autoimmune hepatitis, primary biliary cholangitis, and cryptogenic.

**Table 3 cancers-15-00390-t003:** Comparison between DNHC and the other groups in higher-stage HCCs.

	DNHC(*n* = 17)	Other(*n* = 99)	*p*-Value
Male/Female	15: 2	81: 18	0.518
Age (years)	68.6 ± 10.1	69.1 ± 11.7	0.857
Etiology					
HBV infection	3	(17.6)	21	(21.2)	0.473
HCV infection	5	(29.4)	12	(12.1)
Alcohol	5	(29.4)	34	(34.3)
NAFLD	1	(5.9)	7	(7.1)
Others	3	(17.6)	25	(25.3)
Liver status					
Cirrhosis	10	(58.8)	36	(36.4)	0.080
CP-A	8	(80.0)	16	(44.4)	0.152
CP-B	1	(10.0)	14	(38.9)
CP-C	1	(10.0)	6	(16.7)
Histopathology					
Well-differentiated	4	(44.4)	6	(13.9)	0.090
Moderately-differentiated	4	(44.4)	31	(72.2)
Poorly-differentiated	1	(11.2)	6	(13.9)
BCLC staging					
B	10	(58.8)	42	(42.4)	
C	6	(35.3)	48	(48.5)	0.453
D	1	(5.9)	9	(9.1)	
Up to 7 criteria			
IN: OUT	9 (52.9):8 (47.1)	32 (32.3):67 (67.7)	0.100
Kinki criteria			
B1	5	11	
B2	5	29	0.299
B3	0	2	
TNM staging			
IB, II: III A/B, IVA/B	12:5	41:58	0.026
Vascular invasion	6 (35.3)	45 (45.5)	0.436
First treatment					
Locoregional therapy	14	(82.4)	52	(52.5)	
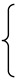	RFA	4	(23.5)	11	(11.1)	
Resection	6	(35.3)	25	(25.3)	
TACE	4	(23.5)	15	(15.2)	
SBRT	0	(0.0)	1	(1.0)	
	Others	3	(17.6)	47	(47.5)	0.022 **
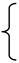	HAIC	0	(0.0)	23	(23.2)	
Systemic *	1	(5.9)	16	(16.2)	
BSC	2	(11.8)	8	(8.1)	
CR to the first treatment	12	(70.6)	40	(40.4)	0.021

HBV, hepatitis B virus; HCV, hepatitis C virus; CP, Child–Pugh classification; AFP, alpha-fetoprotein; DCP, des-gamma-carboxyprothrombin; DNHC, double negative HCC; HCC, hepatocellular carcinoma; BCLC, Barcelona Clinic Liver Cancer; RFA, radiofrequency ablation; TACE, trans-arterial chemoembolization; HAIC, hepatic artery infusion chemotherapy; RT, radiation therapy; BSC, best supportive care; CR, complete response; Data are expressed as mean ± SD. Numbers in parentheses refer to the percentage of patients. * Including Sorafenib, Lenvatinib, and Atezolizumab + Bevacizumab. ** Comparison between locoregional therapy and others.

**Table 4 cancers-15-00390-t004:** Cox proportional hazards regression analysis of factors that were associated with OS in higher-stage HCC.

Factors	Multivariate Analysis					
	β	SE (β)	z	RR	95%CI	*p*-Value
Age	−0.034	0.011	2.996	0.966	0.945–0.988	0.003
Sex	0.282	0.310	0.911	1.326	0.723–2.432	0.362
Up to 7 criteria IN	−0.944	0.276	3.418	0.389	0.227–0.669	<0.001
BCLC stage	0.459	0.222	2.069	1.583	1.025–2.445	0.039
Child–Pugh scores	−0.116	0.078	1.480	1.123	0.963–1.308	0.139
DNHC	−0.769	0.236	2.162	0.463	0.231–0.931	0.031

BCLC, Barcelona Clinic Liver Cancer, DNHC, double negative HCC; DNHC, double-negative HCC.; OS, overall survival; RR, relative risk; SE, standard error; CI, confidence interval.

## Data Availability

No new data were created or analyzed in this study; data sharing is not applicable to this article.

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
