# Peer review of "The Characteristics and Prognosis of Alpha-Fetoprotein and Des-Gamma-Carboxy Prothrombin Double-Negative Hepatocellular Carcinoma at Baseline in Higher BCLC Stages"

_cancers, 2023, doi:10.3390/cancers15020390_

Round 1

Reviewer 1 Report

 The manuscript is clear and well written. It accurately sets out the characteristics and prognosis of AFP/DCP double-negative HCC (DNHC), pointing out that even the negativity of tumor markers could have important therapeutic and prognostic implications, driving to an increased survival rate for high stage DNHC if appropriately treated.

I suggest a rearrangement in figures numbering, because there are two figures numbered "3" (at page 8 and 9).

I encourage the authors to continue the analysis of DNHC with future prospective studies on a larger scale sample of patients.

Author Response

point-by-point response to the reviewer’s comments

Reviewer1

  1. I suggest a rearrangement in figures numbering, because there are two figures numbered "3" (at page 8 and 9).

Reply: Thank you for indicating the mistake. We fixed it.

I encourage the authors to continue the analysis of DNHC with future prospective studies on a larger scale sample of patients.

Reply: Thank you for encouraging us. We will conduct a prospective study.

me references within 5 years.

Reviewer 2 Report

This is a well-written article that identifies an important gap in knowledge.
Abstract:

To modify the expression regarding the prospective enrollment of patients, considering that
the study is retrospective observational.

Discussion

I recommend increasing the number of references published in the last 5 years, the
introduction to discussions of the EASL, AASLD, Asia-Pacific Clinical Practice Guidelines
and the possible links between the level of biomarkers and viral exposure signature.

References:

Only 11 references out of 46 are published in the last 5 years

authors in reference 5

reference 12 does not correspond to the volume

reference 19 does not correspond to the title

reference 45 does not mention the pages

Native English LANGUAGE editing

Author Response

point-by-point response to the reviewer’s comments

Reviewer2

  1. Abstract: To modify the expression regarding the prospective enrollment of patients, considering that the study is retrospective observational.

Reply: Thank you for indicating the mistake. We fixed it to “retrospectively.” 

  1. Discussion: I recommend increasing the number of references published in the last 5 years, the introduction to discussions of the EASL, AASLD, Asia-Pacific Clinical Practice Guidelines and the possible links between the level of biomarkers and viral exposure signature.

Reply: Thank you for your supportive comments. We added the sentences in the discussion section, “Nakamura et al. indicated that the ROC area of DCP was significantly larger than that of AFP in tumors greater than 5 cm in diameter [45].” and “In the clinical practice guidelines of HCC by the European Association for the Study of the Liver (EASL), the American Association for the Study of Liver Diseases (AASLD), and the Asian Pacific Association for the Study of the Liver (APASL), biomarkers were mainly discussed in the vein of early detection [47-49]. The prognostic aspect of the biomarkers was not mentioned. These guidelines indicated the limited ability of the AFP for early detection of HCC and discussed how to increase the accuracy for detection. They also introduced some novel biomarkers, such as glypican 3 [50], Golgi protein 73 [51], osteopontin [52], circulating cell-free DNA [53], and microRNAs [54]. These markers are under active investigation; however, none have been approved for clinical use. These novel biomarkers might be useful for DNHC in higher-stage HCC. The difficulty in defining biomarkers specific for HCC cells is due to its complex genomic landscape with extensive intratumor and inter-tumor heterogeneities. Meanwhile, an emerging concept is that an interplay between viral infection and host genetic background is crucial for maintaining virome homeostasis or causing human disease [55]. Lui et al. demonstrate how viral infection history, obtained using human blood samples and VirScan analysis of antiviral antibodies, can be used to detect HCC in at-risk patients prior to clinical cancer diagnoses [56]. However, it cannot be applied to increasing non-viral HCC. The challenge of biomarker discovery continues.” 

  1. References: Only 11 references out of 46 are published in the last 5 years
  • authors in reference 5
  • reference 12 does not correspond to the volume
  • reference 19 does not correspond to the title
  • reference 45 does not mention the pages

Reply: Thank you for indicating mistakes in the reference. We thoroughly checked the references and fixed them, and added some references within 5 years.